# Social Relationships, Age and the Use of Preventive Health Services: Findings from the German Ageing Survey

**DOI:** 10.3390/ijerph16214272

**Published:** 2019-11-04

**Authors:** Daniel Bremer, Daniel Lüdecke, Olaf von dem Knesebeck

**Affiliations:** 1Department of Medical Psychology & Center for Health Care Research, University Medical Center Hamburg-Eppendorf, 20246 Hamburg, Germany; 2Department of Medical Sociology, University Medical Center Hamburg-Eppendorf, 20246 Hamburg, Germany; d.luedecke@uke.de (D.L.); o.knesebeck@uke.de (O.v.d.K.)

**Keywords:** preventive health services, cancer screening, flu vaccination, social relationships, social networks, informational support, partner, age

## Abstract

This paper investigates the associations between social relationships, age and the use of preventive health services among German adults. Data stem from the German Ageing Survey (10,324 respondents). The use of preventive health services was assessed by asking for regular use of flu vaccination and cancer screening in the past years. Predictors of interest were structural (having a partner, size of the social network) and functional aspects of social relationships (perceived informational support) and age. Logistic regression models were used to measure the associations between preventive health services use and these predictors. Self-perceived health, gender and education were considered as covariates. Having a partner (OR = 1.20, 95% CI: 1.07–1.34) and perceived informational support (OR = 1.38, 95% CI: 1.13–1.69) were associated with a higher probability of getting flu vaccination regularly over the past years. Informational support (OR = 1.42, 95% CI: 1.17–1.72) and having a partner (OR = 1.57, 95% CI: 1.41–1.75) were positively associated with regular cancer screening over the past years. Associations between the size of the social network and use of preventive health services were not statistically significant. Associations between the use of preventive health services and social relationships varied by age. Structural and functional aspects of social relationships may support preventive health behavior. To increase preventive health behavior and the use of preventive health services, it is necessary to integrate information on social relationships into routine care and to strengthen sources of social support.

## 1. Introduction

One of the tasks of the statutory health insurance in Germany is to prevent diseases and to promote health and healthy ageing regardless of peoples’ gender and social status [1]. Ageing populations with changing health needs and chronic conditions are associated with a rising demand for health services [2]. Some disorders can be prevented or influenced positively by preventive interventions [3]. Primary prevention aims at reducing the risk of the onset of a disease. Flu vaccinations, for example, are part of primary prevention, since they have the capability to obviate different serious infectious diseases and to prevent unnecessary hospitalizations and premature deaths [3,4]. Secondary prevention aims at detecting and treating diseases as early as possible. In the case of cancer, screening is especially important, since cancer is responsible for a vast number of deaths worldwide [5].

Following the ‘Behavioral Model of Health Services Use’ by Andersen, use of health services, a measure for health care access, is connected to a multitude of predisposing (e.g., age and gender), enabling (e.g., education, income and social status) and need characteristics (e.g., disease, symptoms and pain) [6]. Various studies showed that Andersen’s framework also applies to the use of preventive health services [7,8,9,10]. For cancer screenings, different patterns of usage could be identified depending on sociodemographic features [11,12,13,14,15], health needs [16,17,18] and socioeconomic or psychological factors [11,14,19,20,21,22,23,24,25,26]. The use of flu vaccinations also varies by sociodemographic factors [27,28], health status [4,28] and socioeconomic characteristics [29,30,31]. Since preventive health services are recommended by health authorities and paid for certain ages and risk groups by the statutory health insurance in Germany, age (as a predisposing characteristic) plays an important role regarding the use of cancer screenings and flu vaccinations [6]. From the age of 50, for example, the statutory health insurance pays for mammography screenings every two years [32]. For men, reaching the age of 45, the yearly use of prostate screenings is covered by the statutory health insurance [32]. Statutory health insurance also covers colorectal cancer screening starting with 50 years of age for both sexes [32]. Flu vaccination is a paid service for everybody and recommended for individuals aged 60 and older. Consequently, health institutions are interested in higher rates of preventive health services use reaching certain ages. In Germany, the Federal Joint Committee (G-BA) decides which health and preventive health services (e.g., early detection programs on cancer) are covered by the statutory health insurance [33]. “In its assessments, the G-BA examines patient benefit […]. Patient benefit is defined as recovery, relief from pain or discomfort, improvement in quality of life, extension of life, or reduction of side effects” [33]. The age limits of preventive health services are part of the G-BA assessment. 

Within the Behavioral Model of Andersen, enabling resources (e.g., education and family) can play a supportive role in the sense of creating potential access and foster realized access to health services [6]. Enabling resources also include the social environment, such as social relationships, that are known to be connected to health, health promotion and the use of health services [6,34,35,36]. Social relationships are characterized by the individuals’ social support, social influence, social engagement and attachment, and have an impact on how they access resources [37]. In addition, international studies have shown that social relationships have a considerable influence on morbidity and mortality [38,39]. Social relationships comprise structural and functional elements [38]. Structural aspects are assessed by quantitative measures (e.g., living arrangements, social network size and frequency of social participation). Functional aspects include elements of financial, instrumental, informational or emotional support. 

To date, few international studies investigated the use of preventive health services in conjunction with dimensions of social relationships [40,41,42,43,44,45]. However, it is not clear how structural and functional aspects of social relationships are linked to preventive health services. Furthermore, it has not been investigated to which extent social relationships do have an impact on the link between age and preventive health services. Therefore, this paper investigates the associations between social relationships, age and the use of preventive health services among German adults aged forty years and older. 

## 2. Materials and Methods 

### 2.1. Data

Data stem from the public release of the fifth wave, in 2014, of the German Ageing Survey (DEAS), provided by the Research Data Centre of the German Centre of Gerontology (DZA) and funded by the Federal Ministry for Family Affairs, Senior Citizens, Women and Youth (BMFSFJ) [46,47]. The population-based survey started in 1996 and included and was representative for individuals 40 years and older in Germany. After the initial survey, other waves followed in 2002, 2008, 2011 and 2014. The interviews covered information on health, occupational status, income, social relationships, life events, psychological well-being and much more [48]. In the fifth wave (2014) 7952 individuals filled out the ‘drop-off’ questionnaire, a questionnaire which was filled by the respondents without an interviewer. Due to panel attrition, each wave introduced new respondents to ‘refresh’ and to stabilize the absolute number of respondents in the sample. As such, 4295 individuals are part of this so-called refreshers sample (54%). The drop-off questionnaire contained items on the use preventive health services (cancer screening and flu vaccination). The response rate of the sample was 61% in 2014. These rates are comparable to other surveys executed in Germany [49]. Our analyses are based on the fifth wave of the German Ageing Survey which included cross-sectional data on perceived informational support, having a partner and social network size for a representative sample of the middle-aged and older population of Germany [49]. A written informed consent was given by every survey participant prior to the interview. The survey respected the Declaration of Helsinki [50]. 

### 2.2. Measures

The use of preventive health services was assessed by asking for regular use of flu vaccination and cancer screening in the past years (no, yes). The predictors in focus were structural (having a partner, size of the social network) and functional aspects of social relationships (perceived informational support) and age. Having a partner was dichotomized (0 ‘having no partner’, 1 ‘having a partner’). ‘Having no partner’ includes singles, divorced, widowed and separated individuals. ‘Having a partner’ is defined by married people and registered partnerships living together. Size of the social network was assessed by asking for ‘people who are important to you and who you maintain regular contact with’. ‘These can include co-workers, neighbors, friends, acquaintances, relatives, and members of your household. Which people are important to you?’ (the respondents could name the people; the names were counted and coded as 0 ‘no one’ to 9 ‘nine or more people’). Perceived informational support was measured by asking ‘When you have important personal decisions to make, do you have anyone you can ask for advice?’ (0 ‘no’, 1 ‘yes’). Age was measured in years. 

Health indicators and other socio-demographic factors were included as covariates. In the current study, self-perceived health (‘How would you rate your present state of health?’) was measured on a five-point scale (1 ‘very good’, 2 ‘good’, 3 ‘average’, 4 ‘bad’ and 5 ‘very bad’) as one health indicator. Furthermore, we included information on pre-existing diseases by taking into account the number of diseases (‘Which of the following diseases and health problems do you have?’). The list of diseases covered fourteen chronic, somatic illnesses, for example, cardiac and circulatory disorders, respiratory problems/ asthma/ shortness of breath, cancer or diabetes. The respondent’s sex was coded by male (= 0) and female (= 1). Education was based on the International Standard Classification of Education (ISCED 1997) and ranged from 1 to 3 (low to higher education). Low education is defined by ISCED 0–2 (= respondents without formal vocational qualification). Medium education based on ISCED 3–4 (= respondents with vocational training including respondents with higher general school certificate without professional training. Higher education represented ISCED 5–6 (= respondents with completed university studies or with completed professional development training).

### 2.3. Analyses

Since the two dependent variables “use of preventive health services” (flu vaccination and cancer screening) are binary (no/yes), two logistic regression models were used to measure the associations between each type of preventive health services use and the predictors (flu vaccination = model 1; cancer screening = model 2). To adjust for disproportional stratifications of the baseline sample and selective panel mortality, weights were used [46,51,52]. To analyze a potential moderation of social relationships on the association between age and the two types of preventive health services, two-way interaction terms were introduced [53]. Three interaction terms were calculated for each of the two types of preventive health services: (1) age * informational support (model 1.1 and model 2.1), (2) age * having a partner (model 1.2 and model 2.2) and (3) age * social network size (model 1.3 and 2.3). In terms of cancer screening, age is added as cubic term to the model, since the relationship between the probability of using cancer screening and age was found to be non-linear. We defined *p* < 0.05 as threshold whether an association was considered statistically significant or not. The analyses were performed with Stata 12 [54] and were replicated with R 3.6.1 [55]. Marginal effects plots were created using the ggeffects-package 0.12.0 [56]. 

## 3. Results

Table 1 shows that 42.5% of the respondents used flu vaccinations regularly in the past years, and 63.3% used cancer screenings. Additionally, 69.9% of the respondents had a partner and 93% reported informational support. On average, social networks included 5.2 important persons with regular contact. More than half of the participants were female and the mean age was 64.5 years. Results also show that 51.6% attained a higher education (ISCED-1997 Coding) and 53.7% reported a good or very good health. On average, respondents reported 2.6 physical diseases. 

### 3.1. Associations between Social Relationships, Age and Preventive Health Services Use

Having a partner (OR = 1.20, 95% CI: 1.07–1.34) and perceived informational support (OR = 1.38, 95% CI: 1.13–1.69) were associated with a higher probability of getting flu vaccination regularly over the past years (Table 2). There was no statistically significant association between the size of the social network and flu vaccination. The probability of using flu vaccination increased by age (OR = 1.06, 95% CI: 1.05–1.06). Reporting a very good (OR = 0.50, 95% CI: 0.40–0.62) or good (OR = 0.73, 95% CI: 0.65–0.82) health was associated with a lower probability of getting flu vaccination regularly, whereas a very bad self-perceived health (OR = 1.70, 95% CI: 1.20–2.40) was connected to a higher probability. Furthermore, the probability of using flu vaccination increased by the number of reported physical diseases (OR = 1.08, 95% CI: 1.04–1.11).

Respondents with perceived informational support (OR = 1.42, 95% CI: 1.17–1.72) and having a partner (OR = 1.57, 95% CI: 1.41–1.75) were more likely to use cancer screening (Table 2). The odds ratios of the size of the social network and of education on using cancer screenings were not statistically significant. Furthermore, we found a statistically significant relationship between age and the use of cancer screening (age: OR = 1.34, 95% CI: 1.29–1.40; age 2: OR = 1.00, 95% CI: 1.00–1.00). Age was positively associated with the use of cancer screening up to around 63 years. However, ageing 63 and older the association is negative. Individuals with a higher education (OR = 1.14, 95% CI: 1.02–1.27) were more likely to utilize cancer screening. Moreover, reporting a very good health (OR = 0.78, 95% CI: 0.64–0.95) or a bad health status (OR = 0.78, 95% CI: 0.65–0.94) was associated with a lower probability of getting cancer screenings regularly. Furthermore, the probability of using cancer screening slightly increased by the number of reported physical diseases (OR = 1.05, 95% CI: 1.01–1.08). 

### 3.2. Moderation of Social Relationships on Age and Preventive Health Services Use

The associations between the use of preventive health services (flu vaccination, cancer screening) and age varied by social relationships (having a partner, perceived informational support). The proportion of respondents using flu vaccination increased by age. We found a difference if someone perceived informational support or not for the age group of 60 to 75 years old (Figure 1a). Within that age group, respondents who perceived informational support showed a significantly higher chance of getting flu shots. The proportion of people using cancer screening increased within the age group of 40 to 65 years, then decreasing constantly until the age of 95 (Figure 1b). The chance of using cancer screenings is significantly higher for people aged 43 to 66 if they perceived informational support. 

Figure 2a shows the use of flu vaccination by age and having a partner. People having a partner only had a slightly higher chance of using flu shots than people without a partner. In both groups, the proportion of demanding flu vaccination increased by age. For cancer screenings, Figure 2b shows a different picture. From 50 up to 95 years, the ratio of people using cancer screening was higher if respondents reported having a partner. The highest proportion of cancer screening users could be measured at 65 years if a partner was present (75%) and at around 60 years if not having a partner (65%).

Figure 3a,b do not show any major differences in using preventive health services (flu vaccination and cancer screening) taking age and the size of the social network into account. Merely, the use of cancer screenings demonstrated some small gaps between the subgroups (of social network size) in certain age intervals (Figure 3b). The general curve characteristics in Figure 3a,b were similar to the figures above. 

The figures, shown above, based on interaction models which are presented in the Appendix A.

## 4. Discussion

### 4.1. Summary

This study revealed that a functional aspect of social relationships (perceived informational support) was associated with a higher probability of using flu vaccination and cancer screening in a German sample of people aged 40 years and older. The association between one structural factor of social relationships and the use of preventive health services was statistically significant (having a partner), the other one was not (size of the social network). Moreover, the probability of using flu vaccination increased by age. Considering the use of cancer screening, the odds were increasing by age among those aged 40 to 63. After the age of 63 people were less likely to use cancer screening. Potentially, this observation could be partly explained by recommended age limits with regard to certain cancer screening types in Germany (e.g., colon, rectum and mammography screening) [57]. Furthermore, the associations between the use of preventive health services and age varied by different dimensions of social relationships. With regards to the use of flu vaccination, perceiving informal support seems to be a supportive factor especially for the age of 60 to 75. This held also true for people in their early 40s up to 66 with regard to cancer screening. Having a partner seemed to encourage the use of cancer screening, especially for people aged 50 years and older. 

Kinney, Bloor, Martin et al. reported that people who were structurally well integrated, had a higher chance of reporting recent use of colorectal cancer screening [41]. Functional and instrumental support, representing functional aspects of social relationships, were not associated with the use of colorectal cancer screening. While the findings on the positive association between social relationships and cancer screening were in line with our results, we also found statistically significant associations between functional aspects of social ties and preventive health services. Allen, Sorensen, Stoddard et al. investigated the relationship between social network characteristics and breast cancer screening among employed women [40]. In their multivariable analyses, social network characteristics did not predict using regular screening. Only the perception that screening is socially desirable led to increased usage. Potentially, our results on social network size could be in line with these findings. Suarez, Ramirez, Villareal et al. formed an index on social integration including structural and functional elements of social relationships (number of close relatives and friends, frequency of contact, church membership) and linked it to various types of cancer screenings among four U. S. Hispanic groups (Mexican, Central-American, Cuban and Puerto Rican) [42]. Their results showed a complex picture of no, weak and strong associations depending on the type of screening and the four Hispanic groups.

Like other studies, we found that age was associated with vaccination uptake [58,59]. Being married or living with others has been associated with vaccination acceptance in some studies [60,61]. Furthermore, several studies on barriers and facilitators of getting influenza immunization indicated that advice from and health discussions with family and friends may trigger the acceptance and use of flu vaccination [43,44,45,62]. Consequently, our results concerning the positive associations between functional aspects of social relationship could support and add to the existing literature on social ties and the use of flu vaccination. 

All in all, functional and structural aspects of social relationships were associated with a higher probability of using preventive health services. Living in a partnership and perceiving informational support seem to enable individuals to access preventive health services and to support their preventive health behavior. Furthermore, the results showed that age played a crucial role in using preventive health services. In the age curves of preventive health services, fundamental differences between flu vaccination and cancer screening could be shown. While the age curve of flu vaccination almost showed a linear trend, the age curve of cancer screening was concave. The moderator analyses showed that social relationships moderate the link between age and the use of preventive health services. In the case of flu vaccination, individuals, aged 60–75 and perceiving informational support, had a higher chance of use. With regard to cancer screening, informational support increased the probability of use in the age group 43–66 and living in a partnership promoted the chance of use among those 50 years and older. Consequently, functional and structural aspects of social relationships seem to have the potential to enable the use of preventive health services, especially of cancer screening.

### 4.2. Limitations

Methodological limitations need to be taken into account, when interpreting the results. Due to changes of the measurement of preventive health services between the waves of the German Ageing Survey, only cross-sectional data were used for the analyses [48]. Therefore, it is not possible to comment on changes over time and causal directions. Secondly, the items on using preventive health services were based on self-reports and on a rather vague time span by asking for regular use of flu vaccination and cancer screening in the past years. The time span covering the preventive health services can be quite long, and considering the older age of some respondents, risk of memory bias could be existent regarding the use of preventive health services [63]. Moreover, the item on using cancer screening did not specify which type of cancer screening was meant. It was formulated in general terms. Consequently, further subgroup analyses were not possible. Moreover, the German Ageing Survey did not provide information on the motives for using preventive health services, their quality and adequacy. Consequently, our preventive health services item represents a proxy for “realized access” [6] only. 

Besides methodological limitations, there is an ongoing debate on the effectiveness or harmfulness of preventive health services, especially, concerning cancer screenings and flu vaccination [64,65,66]. It is important to keep that in mind, when discussing the use of preventive health services in general. Simonsen et al. questioned the effectiveness of flu vaccination, for example, concerning mortality benefits of flu vaccination in elderly people, since frailty selection bias, the use of non-specific endpoints could have resulted in exaggerating vaccine benefits in cohort studies [66]. They conclude that “the remaining evidence base is currently insufficient to indicate the magnitude of the mortality benefit, if any, that elderly people derive from the vaccination programme” [66]. Furthermore, flu vaccination may have side-effects for health. Kwok stated that “vaccines do carry risks, ranging from rashes or tenderness at the site of injection to fever-associated seizures […] and dangerous infections in those with compromised immune systems” [67], although severe complications are unusual and it is difficult to show that a vaccine is the cause for them [67]. With regard to the controversy over cancer screenings, radiation risks are one part of it [64]. The controversy also includes arguments on over-treatment and over-diagnosis of cancer. Esserman et al. noted that “screening and patient awareness have increased the chance of identifying a spectrum of cancers, some of which are not life threatening. Policies that prevent or reduce the chance of overdiagnosis and avoid overtreatment are needed, while maintaining those gains by which early detection is a major contributor to decreasing mortality and locally advanced disease” [68].

Concepts of social relationships which were used in our study (having a partner, informational support and social network size) were only indirect measures of structural (having a partner, social network size) and functional aspects (informational support) of social relationships. Especially, the partner variable or the size of social networks were only rough measures for social connectedness and the feeling for belonging and being cared for. Our data did not include information on the qualitative partnership or social network functioning which could be differentiated into costs (e.g., psychological distress, destructive conflicts) and benefits (e.g., belonging, meaning). Since social relationships could have positive and negative aspects, they could lead to different health and health behavior outcomes [69]. Although the indirect approach (referring on socio-demographic proxies) and the direct approach (linking meaningfulness and importance to social relations) are used in the German Ageing Survey, specific information about the quality of social relationships and the level of social support were missing [70]. 

## 5. Conclusions

Having a partner and perceived informational support were associated with a higher probability of using preventive health services. The social environment, like structural and functional aspects of social relationships, may support preventive health behavior, especially within certain age groups (flu vaccination: informational support and age of 60–75; cancer screening: informational support and age of 43–66, having a partner and age of 50–95). If health policy and health professionals want to increase preventive health behavior and the use of preventive health services, it is necessary to integrate information on social relationships into routine care and to strengthen sources of social support. 

## Figures and Tables

**Figure 1 ijerph-16-04272-f001:**
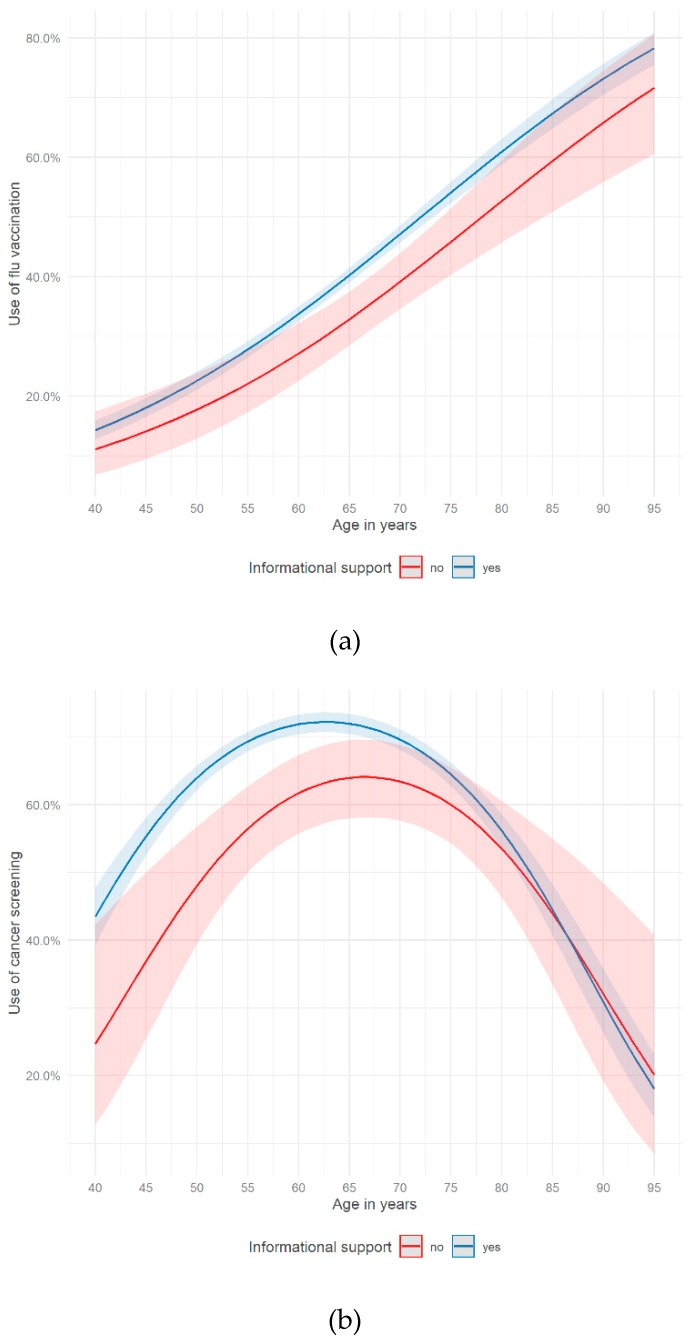
Use of (**a**) flu vaccination (Model 1.1) and (**b**) cancer screening (Model 2.1) on age and informational support (German Ageing Survey, 2014).

**Figure 2 ijerph-16-04272-f002:**
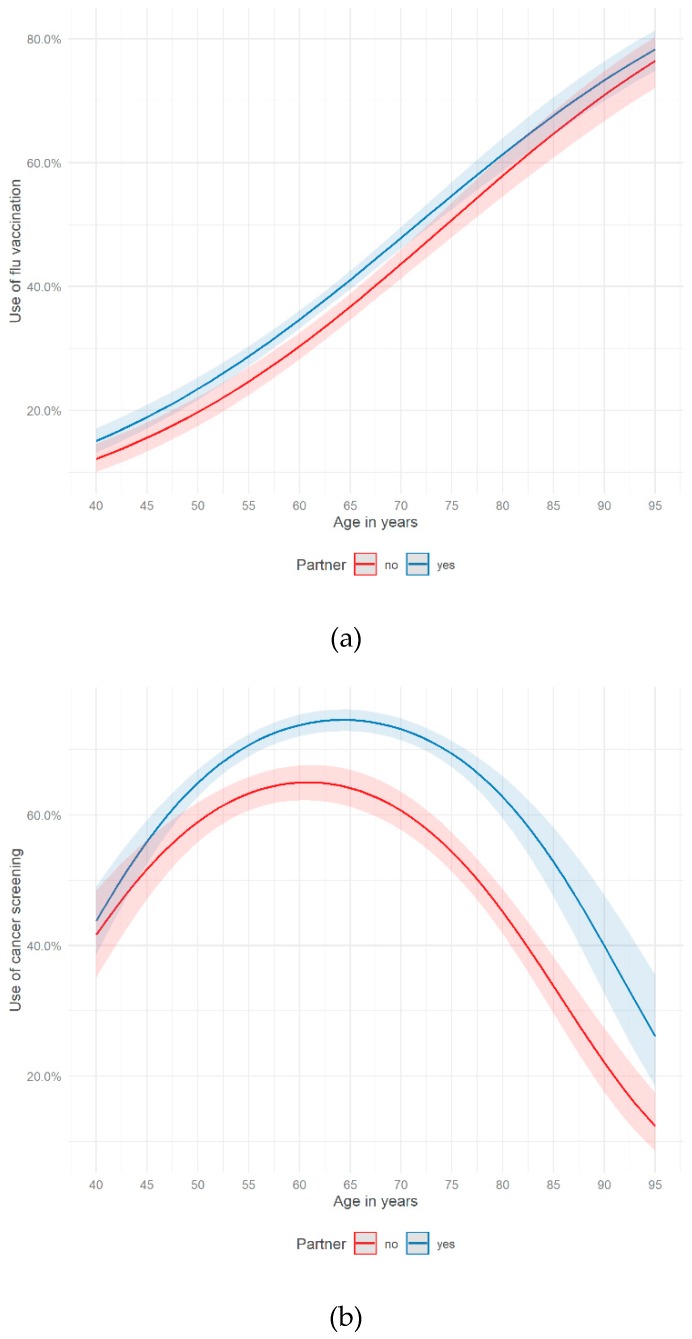
Use of (**a**) flu vaccination (Model 1.2.) and (**b**) cancer screening (Model 2.2.) on age and having a partner (German Ageing Survey, 2014).

**Figure 3 ijerph-16-04272-f003:**
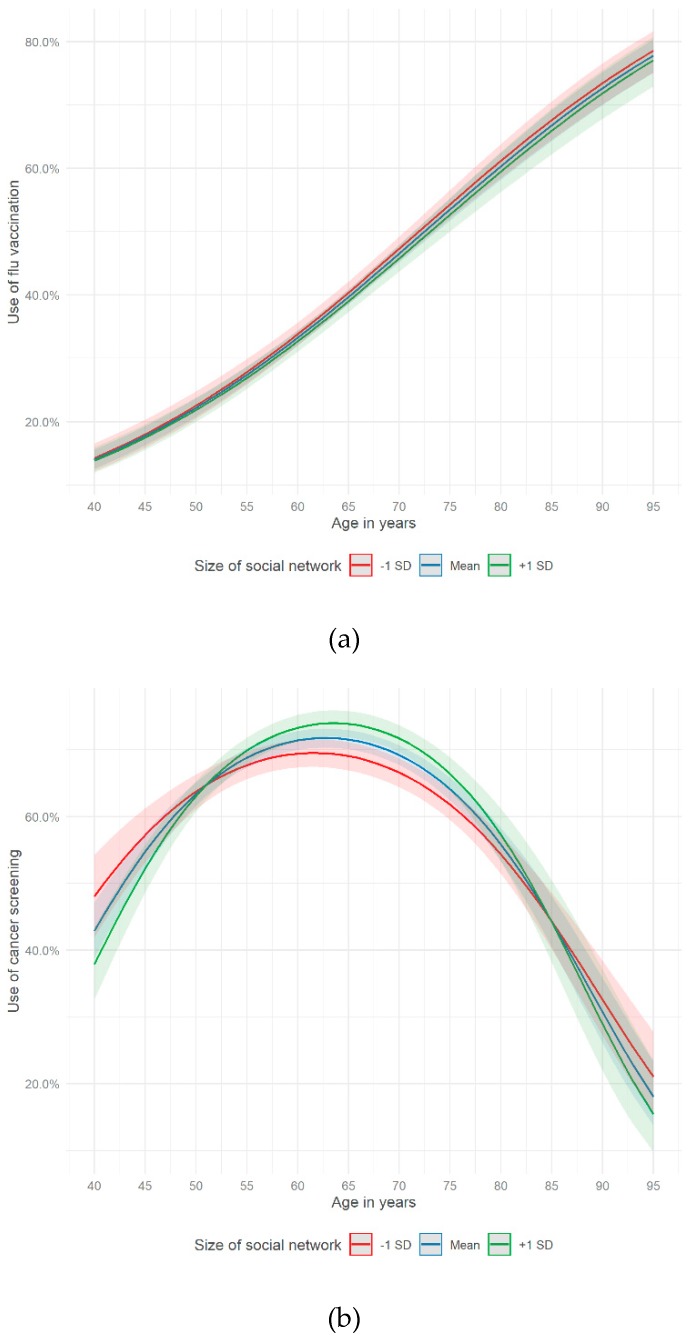
Use of (**a**) flu vaccination (Model 1.3) and (**b**) cancer screening (Model 2.3) on age and social network size (German Ageing Survey, 2014).

**Table 1 ijerph-16-04272-t001:** Descriptive statistics of the sample by drop-off questionnaire (*n* = 7952, (German Ageing Survey), 2014).

Variables	*N* (%), Mean (SD)
Female: *N* (%)	4056 (51.01)
Age: Mean (SD)	64.54 (11.24)
Education ^a^ (ISCED-1997 Coding): *N* (%)	
*ISCED-1: low*	521 (6.55)
*ISCED-2: medium*	4100 (51.56)
*ISCED-3: high*	3329 (41.86)
Self-perceived health ^b^: *N* (%)	
*Very good*	641 (8.06)
*Good*	3631 (45.66)
*Average*	2857 (35.93)
*Bad*	670 (8.43)
*Very bad*	145 (1.82)
**Number of physical diseases** ^**c**^ **: Mean (SD)**	2.6 (1.89)
Having a partner (= yes)^d^: *N* (%)	5556 (69.87)
Social network size: Mean (SD)	5.22 (2.70)
Perceived informational support (= yes) ^e^: *N* (%)	7396 (93.01)
Regular flu vaccination in the past years (= yes) ^f^: *N* (%)	3383 (42.54)
Regular cancer screening in the past years (= yes) ^g^: *N* (%)	5034 (63.30)
*Missing values (out of 7952): ^a^ 1, ^b^ 8, ^c^ 148, ^d^ 16, ^e^ 22, ^f^ 202, ^g^ 279*	

^a^: 1 missing value (mv), ^b^: 8 mv, ^c^: 148 mv, ^d^: 16 mv, ^e^: 22 mv, ^f^: 202 mv, ^g^: 279 mv.

**Table 2 ijerph-16-04272-t002:** Logistic regression models for flu vaccination (Model 1) and cancer screening (Model 2) (German Ageing Survey, 2014).

Variables	Seasonal Flu Vaccination (Model 1)		Cancer Screening (Model 2)	
Predictors	Odds Ratio	95% CI	Odds Ratio	95% CI
Partner (Ref. no): yes	1.20	1.07–1.34	1.57	1.41–1.75
Social network size (number of important persons with regular contact)	0.99	0.97–1.01	1.02	1.00–1.04
Informational support (Ref. no): yes	1.38	1.13–1.69	1.42	1.17–1.72
Gender (Ref. male): female	1.03	0.93–1.15	2.38	2.14–2.64
Age in years	1.06	1.05–1.06	1.34	1.29–1.40
Age in years (cubic term)			1.00	1.00–1.00
Education (ISCED-1997) (Ref. ISCED-2: medium)				
ISCED-1: low	0.89	0.75–1.06	0.87	0.73–1.04
ISCED-3: high	1.03	0.93–1.15	1.14	1.02–1.27
Self-perceived health (Ref. average)				
Very good	0.50	0.40–0.62	0.78	0.64–0.95
Good	0.73	0.65–0.82	1.07	0.95–1.20
Bad	1.07	0.89–1.29	0.78	0.65–0.94
Very bad	1.70	1.20–2.40	0.77	0.55–1.08
Number of physical diseases	1.08	1.04–1.11	1.05	1.01–1.08
Intercept	0.01	0.01–0.02	0.00	0.00–0.00
Number of observations	7588		7515	

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
