# Peer review of "Social Relationships, Age and the Use of Preventive Health Services: Findings from the German Ageing Survey"

_ijerph, 2019, doi:10.3390/ijerph16214272_

Round 1

Reviewer 1 Report

The abstract is almost uninformative. In the results section provide relevant figures (OR and 95%CIs) In the introduction section the authors stated "....cancer screenings can be defined as primary and secondary prevention 36 depending on the (preliminary or early) stage of cancer". This is clearly a mistake, even if referenced, since cancer screening is a secondary prevention only. Again in the Introduction section, it is reported that "Men are able to use prostate screenings on a yearly basis reaching the age of 45". It is not clear the Evidence base of this choice, since there isa lot of controversy about using the PSA test to look for prostate cancer in men. As an example, in the USA, the Preventive Services Task Force had concluded that the potential risks of PSA screening in healthy men outweigh the potential benefits. Give a reasonable explanation of this choice in Germany. The measure of regular use of cancer screening is poor. Which type of cancer are the authors speaking for? If the question present in the survey is too generic, without a specification of the type of screening, as the authors stated in the limitations, this must be as a tool for changing the future survey In the paragraph "Analysis", indicate the type of model used. Moreover, indicate also the H-L test. In table 2, eliminate the p-value columns, the 95% CIs are sufficiently informative. Moreover, indicate the reference group for Gender. Indicate "seasonal flu vaccination" and not simply "flu vaccination"

Reviewer 2 Report

Thank you for the opportunity to review this research that I found particularly interesting.

I provide here some comments which could potentially improve the first version of the manuscript.

Main comments

1/ The section “Data” is pretty unclear to me. I am not a specialist of survey data but it might also be the case of some of your readers, and the terms “drop-off questionnaires” and “refreshers sample” are not explicit. It is also unclear from this section what share of the German population over 40 was included in your survey and if it is expected to be representative of the whole population of interest? Finally, I would first state that your analyses are based on the fifth wave, before giving the number of individuals in this wave.

2/ Your outcomes of interest are quite common in your population so odds ratio might convey an inflated effect size, and be better replaced by relative rates (see for instance: https://www.ncbi.nlm.nih.gov/pubmed/12746247). You might want to discuss your methodological choice.

3/ In the last paragraph of the measures section of the method, you mention a health indicator. Which one is it? Is it the self-perceived health that you mention in the next sentence? Please clarify.

4/ In the Analyses part of the methods section, you mention the dependent variable “use of preventive health services” but in reality, when looking at the findings, we see that there are two distinct dependent variables (flu vaccination and cancer screening). Please clarify/modify.

5/ In Table 1, start first by the variables which describe your population, and then by your outcomes of interest.

6/ The section 3.1 of the results section needs to be restructured keeping in mind the primary objectives of your research. First, report associations between social support and your dependent variables, rather than starting by the adjustors which are not your main focus of interest.

7/ You find that cancer screening decreases in the oldest populations. In some countries, organized cancer screening is only recommended up until an age limit (for example organized breast cancer screening in France is only recommended up to 74 years old). The fact that guidelines for cancer screening are not as strong in the oldest populations might partly explain your findings and it would be worth mentioning it in the discussion.

8/ “Besides methodological limitations, there is an ongoing debate on the effectiveness or harmfulness of PHS, especially, concerning cancer screenings and flu vaccination [51-53]”. Can you be more explicit, for instance over-diagnosis of cancers which would not have evolved, side effects and lack of effectiveness for flu vaccines?

More minor comments

1/ A lot of references are in German. I think it might not be the best when addressing an international audience, especially when such references are not used to describe the German context. For instance: “For cancer screenings, different patterns of usage could be identified depending on sociodemographic features [11, 12], health needs [13, 14] and socioeconomic or psychological factors [11, 15-17].” => A lot of these references are in German, while there are also research published in the international literature on these issues.

2/ The use of abbreviation can sometimes be useful to decrease “heaviness” of a text but I think here it is more confusing than helpful: SR, PHS, etc. It’s a lot of them and not so common and well-known abbreviations. You might want to modify it.

3/ Some English language rewording could be necessary or a global typo check. For example, line 53 “starting with the 50 years of age”; line 80 “The drop-off questionnaire contained the items on the use PHSs”=> “contained items on the use of PHSs”; line 109: one parenthesis is missing; Line 180: perceived and not “perceive informational support”; line 188: to be a supportive factor; line 190: seemed to encourage not “seems”; line 200: led not “lead”; line 229: “the items of using PHS”: on using PHS.

4/ Some parts are also unclear and need to be rephrased. Line 96 “(the respondents could name the people; 0 “no one” to 9 “nine and more people)”: did they name them or just give the number?; “On the one hand, associations between structural factors of SR and the use of PHS were statistically significant (having a partner), and on the other hand, not”: not fully explicit, just state that some are associated and others are not.

5/ When mentioning the software used in the research, why do you only put the reference for the R software and not the Stata one? Give the versions used for each software and also for the R package you mention. R packages tend to evolve quickly.

6/ I think it is 95% CI and not CI95% (in the text). Also add in the table 1 95% before CI because the boundaries are currently missing. Also under table 1, mention that p-values are in bold when significant.

7/ You don’t mention in the methods what threshold was used to define whether an association was considered significant or not. It should be added.

8/ It is weird to put the references just after the names of authors and not at the end of the related sentence in for instance: “Kinney, Bloor, Martin et al. [30] reported that people, who were structurally well integrated, had a higher chance of reporting recent use of colorectal cancer screening. Functional and instrumental support, representing functional aspects of SR,were not associated with the use of colorectal cancer screening. While the findings on the positive association between SR and cancer screening were in line with our results, we also found statistically significant associations between functional aspects of social ties and PHS. Allen, Sorensen, Stoddard et al. [29] investigated the relationship between social network characteristics and breast cancer screening among employed women.”.

9/ Line 199: “multivariate analyses”: did you mean “multivariable”? See https://www.ncbi.nlm.nih.gov/pmc/articles/PMC3518362/

10/ Line 205: « the Hispanic group »: what is a hispanic group ?

11/ “The time span covering the PHS can be quite long, and considering the older age of some respondents, risk of memory bias could be existent regarding the use of PHS [50].” : can you remind the readers of the maximal time span here so that it is easier to assess this limitation?

12/ “Especially, the partner variable was only a rough measure for social connectedness and the feeling for belonging and being cared for.”: Would not the size of social networks be a better proxy on that aspect?

“Our data did not include information on the partnership functioning which could be differentiated into costs and benefits, or in other words, positive and negative functioning leading to different health and health behavior outcomes [54].”: not very clear, I would advise to rephrase.

Reviewer 3 Report

Thank you for inviting me to review your article, which I enjoyed reading.

I believe, with minor changes this article makes a worthwhile contribution.

Importantly, please could I ask you make comment about whether your participants were know to have or had any pre-existing diseases or disability? These may positively or negatively effect their uptake og PHS.

If this is not included in your study I suggest it may be a limitation?

Round 2

Reviewer 1 Report

The authors made the requrested modifications and now the manuscript is publishable

Reviewer 2 Report

Thank you for taking into account my comments and those of the two other reviewers. Although added at my request, the discussion on the controversy over flu vaccines and cancer screening might be disproportionately long but this is a really minor comment.